# Evaluation of NV556, a Novel Cyclophilin Inhibitor, as a Potential Antifibrotic Compound for Liver Fibrosis

**DOI:** 10.3390/cells8111409

**Published:** 2019-11-08

**Authors:** Sonia Simón Serrano, Alvar Grönberg, Lisa Longato, Krista Rombouts, Joseph Kuo, Matthew Gregory, Steven Moss, Eskil Elmér, Giuseppe Mazza, Philippe Gallay, Massimo Pinzani, Magnus J. Hansson, Ramin Massoumi

**Affiliations:** 1Department of Laboratory Medicine, Translational Cancer Research, Lund University, Medicon Village, 223 63 Lund, Sweden; 2NeuroVive Pharmaceutical AB, 223 63 Lund, Sweden; 3Engitix Ltd., London NW3 2PF, UK; 4Regenerative Medicine & Fibrosis Group, Institute for Liver & Digestive Health, University College London, Royal Free Hospital, London NW3 2PF, UK; 5Department of Immunology & Microbiology, The Scripps Research Institute, La Jolla, CA 92037, USA; 6Isomerase Therapeutics Ltd., Cambridge CB10 1XL, UK; 7Mitochondrial Medicine, Department of Clinical Sciences Lund, Faculty of Medicine, Lund University, 221 84 Lund, Sweden

**Keywords:** cyclophilin, liver fibrosis, NV556, methionine-choline-deficient (MCD) diet, nonalcoholic steatohepatitis, STAM, hepatic stellate cells (HSC), 3D in vitro model, decellularized human liver

## Abstract

Hepatic fibrosis can result as a pathological response to nonalcoholic steatohepatitis (NASH). Cirrhosis, the late stage of fibrosis, has been linked to poor survival and an increased risk of developing hepatocellular carcinoma, with limited treatment options available. Therefore, there is an unmet need for novel effective antifibrotic compounds. Cyclophilins are peptidyl-prolyl cis-trans isomerases that facilitate protein folding and conformational changes affecting the function of the targeted proteins. Due to their activity, cyclophilins have been presented as key factors in several stages of the fibrotic process. In this study, we investigated the antifibrotic effects of NV556, a novel potent sanglifehrin-based cyclophilin inhibitor, in vitro and in vivo. NV556 potential antifibrotic effect was evaluated in two well-established animal models of NASH, STAM, and methionine-choline-deficient (MCD) mice, as well as in an in vitro 3D human liver ECM culture of LX2 cells, a human hepatic stellate cell line. We demonstrate that NV556 decreased liver fibrosis in both STAM and MCD in vivo models and decreased collagen production in TGFβ1-activated hepatic stellate cells in vitro. Taken together, these results present NV556 as a potential candidate for the treatment of liver fibrosis.

## 1. Introduction

Hepatic fibrosis is a pathological response to chronic liver injuries, including chronic alcohol consumption, viral infections such as Hepatitis B (HBV) and C (HCV), or nonalcoholic fatty liver disease (NAFLD) and its severe form, nonalcoholic steatohepatitis (NASH). The progression from NAFLD to NASH is marked by an excessive accumulation of fat in the liver, leading to necrosis and apoptosis of the hepatocytes, which coincide with the formation of liver fibrosis [1]. These events are characterized by inflammation and hepatic stellate cell (HSCs) activation, leading to tissue remodeling and repair with increased accumulation of fibrillar collagens, especially collagen I and III. Cirrhosis, the late stage of fibrosis, has been linked to poor survival and increased risk of developing hepatocellular carcinoma [2]. Currently, liver transplantation is the only curative treatment for patients with cirrhosis and its clinical complications. On the other hand, reversibility of fibrosis has been observed in vivo and in patients when the origin of the chronic injury is resolved, increasing the interest for antifibrotic compounds [3,4,5,6,7,8,9]. Under physiological conditions, HSCs are found in a nonproliferative, quiescent state, whereas liver injury promotes activation and transdifferentiation of HSCs into myofibroblast-like cells, which are the main extracellular matrix (ECM)-producing cells [10,11]. The activation of HSCs is characterized by the expression of α-smooth muscle actin (α-SMA) and type I collagen [4].

Finding a better treatment to counteract or reverse liver fibrosis is one of the greatest challenges in this research field and has promoted an upsurge in interest in the development of new therapies, by making usage of new models for drug discovery [12]. In recent years, the development of well-defined three-dimensional (3D) in vitro models that mimic native tissue ECM structures has demonstrated the importance of cell-matrix biomechanics occurring within a 3D microenvironment in the recapitulation of the complex interactions between HSCs, hepatocytes, and other non-parenchymal cells, in the process of liver fibrogenesis. Along these lines, Mazza G. et al. [13] have developed a novel proprietary technique to rapidly decellularize healthy human liver tissue. This novel technique allows for the isolation of the ECM from human healthy liver tissue by removing the cells but maintaining its ECM properties and structure. The resulting organ-derived bio scaffold has thoroughly been characterized with respect to the preservation of its native ECM, protein composition, 3D microarchitecture, bioactivity, angiogenic potential, topography, and biochemical/ biomechanical properties, compared with the native tissue. The obtained scaffolds have been demonstrated to mimic the interaction between resident liver cells and the surrounding ECM microenvironment and thus play a crucial role in identifying new and more specific therapeutic targets.

Cyclophilins (Cyps) are a family of conserved proteins present in all cell types and organisms. Multiple cyclophilins have been identified in the human genome with differences in structure and cellular location [14,15]. Cyps present peptidyl-prolyl cis-trans isomerase activity (PPIase activity), catalyzing the isomerization of peptides from the cis to trans form, facilitating protein folding. In addition, this PPIase activity can induce conformational changes in the targeted proteins, inducing their activation or inactivation [15]. Previous studies have confirmed the significant upregulation of cyclophilin B (CypB) in fibrotic tissues compared to normal tissues in a CCl_4_-induced liver fibrosis model and Cyclophilin A (CypA) in oral submucous fibrosis, in addition to an increased secretion of CypA in experimental biliary atresia [16,17,18]. Due to their PPIase activity, cyclophilins play an important role at different stages of the fibrotic process, including inflammation, activation of apoptotic pathways in hepatocytes, and activation of the HSCs, leading to increased collagen production [18,19,20,21].

Cyclosporine A (CsA) was the first cyclophilin inhibitor discovered and is used as an immunosuppressant after organ transplantation via its additional inhibitory action on calcineurin [22]. This cyclophilin inhibitor was shown to reduce recurrence of primary biliary cirrhosis (PBC) after liver transplantation more effectively than tacrolimus, a calcineurin inhibitor that does not interact with cyclophilins [23]. A recent non-immunosuppressive cell-impermeable cyclophilin inhibitor, MM284, reduced the myocardial injury and fibrosis in a mouse model of troponin I-induced autoimmune myocarditis [19]. In an experimental biliary atresia model, MM284 treatment decreased infiltration of immune cells in the liver [17]. Treatment with the cyclophilin inhibitor NIM811, a non-immunosuppressive cyclosporin derivative not binding to calcineurin, presented a protective role in hepatocytes by reducing liver necrosis in a bile duct ligation (BDL) experimental model [24]. In a CCl_4_-induced liver fibrosis model, NIM811 treatment significantly decreased the overexpression of CypB and reduced liver necrosis and fibrosis [18]. Supporting these findings, treatment with CRV431, a cyclosporin analog, was also able to decrease liver fibrosis in the CCl_4_ model [25].

In this study, we analyzed the possible antifibrotic effects of NV556, a novel sanglifehrin-based potent cyclophilin inhibitor previously referred to as NVP018 [26]. We demonstrate that NV556 administration was able to decrease liver fibrosis in two distinct and well-established NASH in vivo models. In vitro, NV556 decreased collagen production in the human hepatic stellate cell line LX2 grown in 3D human liver ECM scaffolds.

## 2. Materials and Methods

### 2.1. Reagents

NV556 (NeuroVive Pharmaceutical AB, Lund, Sweden) and FXR agonist, obeticholic acid (MedChemExpress) were dissolved for dosing solution, with the vehicle containing 5% ethanol, 5% Cremophor EL (Kolliphor^®^ EL, Sigma-Aldrich), and 90% saline. Telmisartan (Micardis^®^, Boehringer Ingelheim) was dissolved in pure water. Streptozotocin (STZ) for the induction of NASH in the STAM model was purchased from Sigma-Aldrich, USA.

For the in vitro studies, a stock solution of NV556 was prepared in DMSO, at a concentration of 10 mM. TGF-β1 (R&D systems, Abingdon, UK) was dissolved in 4 mM of HCl containing 1 mg/mL of bovine serum albumin.

### 2.2. Animal Models

#### 2.2.1. Methionine-Choline-Deficient (MCD) Diet

This study was performed at Physiogenex S.A.S, Labège, France. C57BL6/J male mice were purchased from Charles River (Écully, France) at 8 weeks of age, and after 1 week of acclimation (9 weeks of age), mice were fed a methionine-choline-deficient (MCD) diet for 8 weeks (Cat#A02082002B, Research Diets, New Brunswick, NJ, USA). All procedures were performed according to the Guide for the Care and Use of Laboratory Animals (revised 1996 and 2011, 2010/63/EU) and French Laws. After week 1 of the MCD diet (10 weeks of age), mice were weighed and bled from the tail tip, to measure ALT and AST levels using ABX Pentra reagents and Pentra C400 (Horiba Medical). Next, mice were randomly divided into 3 groups (*n* = 10 per group). Each group was treated orally, according to their given treatment: vehicle, NV556 (100 mg/kg), or obeticholic acid (30 mg/kg), for 7 weeks. One mouse in the obeticholic acid group died after 37 days of treatment due to wounding by other mice. Body weight was measured 3 times per week from the start of the MCD diet until the end of the experiment. At weeks 3 and 7 of treatment (13 and 17 weeks of age), mice were sacrificed by cervical dislocation under isoflurane anesthesia and exsanguinated with sterile saline.

#### 2.2.2. STAM Model of Nonalcoholic Steatohepatitis

This model was performed by SMC Laboratories, Inc. (Tokyo, Japan), in 24 C57BL/6 male mice, as previously described by Fujii, Shibazaki [27]. All procedures were performed in accordance with the Japanese Pharmacological Society Guidelines for Animal Use. Briefly, 2 days after birth, NASH was induced in 24 male mice by a single subcutaneous injection of 200 µg of streptozotocin. At 4 weeks of age, mice were changed to an ad libitum high-fat diet (HFD, 57 kcal% fat, Cat#HFD32, CLEA Japan). At 5 weeks of age, 24 mice were randomly divided in 3 groups and treated daily by oral administration, with their respective treatments (10 mL/kg vehicle, 100 mg/kg of NV556 or 5 mg/kg of Telmisartan), up to week 12. Body weight was measured daily during treatment. At 12 weeks of age, mice were sacrificed by exsanguination through direct cardiac puncture under anesthesia.

### 2.3. Liver Histology and Biochemistry Analysis

For the MCD model, at 3 and 7 weeks of experimental phase (13 and 17 weeks of age) blood samples were taken for the analysis of ALT and AST levels. Liver homogenate originated from flash/snap-frozen liver tissue was used for the analysis of hepatic total cholesterol, triglycerides, and fatty acids analysis; the protocol for isolation and quantification was based on Miao, Zondlo [28]. For histological analysis, paraffin-embedded samples were stained with hematoxylin and eosin or Sirius Red, as previously described [29]. Slides were digitalized with the NanoZoomer scanner (Hamamatsu) in bright field conditions (objective ×20). For each individual, a NAFLD scoring system adapted from Kleiner and Brunt [30] was used to perform a semi-quantitative evaluation of NAFLD by analyzing hepatocellular steatosis, liver inflammation, lobular fibrosis, and hepatocyte ballooning.

For the analysis of plasma in the STAM model, non-fasting blood was collected in polypropylene tubes with anticoagulant (Novo-Heparin, Mochida Pharmaceutical) by submandibular bleeding at 6 and 7 weeks of age. Blood glucose levels were measured in whole blood with Life Check. Plasma triglycerides levels were quantified with FUJI DRI-CHEM 7000 (Fujifilm, Tokyo, Japan). Plasma insulin levels were quantified by employing ultra-sensitive Insulin ELISA kit (Morianaga Institute of Biological Science, Yokohama, Japan). Liver total lipid-extracts were collected by Folch’s method [31]. Liver triglyceride content extracts were assessed using the Triglyceride E-test (Wako Pure Chemical industries, Osaka, Japan). For histological analysis, sections were cut from paraffin blocks of liver tissue prefixed in Bouin’s solution and stained with Lillie-Mayer’s Hematoxylin (Muto Pure Chemicals Co., Ltd., Tokyo, Japan) and eosin solution (Wako Pure Chemical Industries). NAS score was adapted from Kleiner and Brunt [30]. For the analysis of collagen deposition, Bouin’s fixed liver sections were stained with picro Sirius red solution (Waldeck GmbH & Co. KG, Münster, Germany). Fibrosis-positive areas were quantified by capturing images around the central vein at 200× fold and the positive areas in 5 fields and measured with ImageJ software (National Institute of Health, Bethesda, MD, USA). For the staining of α-SMA, sections were cut from liver tissues embedded in Tissue-Tek OCT fixed with acetone. Endogenous peroxidase activity was blocked with H_2_O_2_ and incubated with Block Ace (Sumitomo Dainippon Pharma Co. Ltd., Osaka, Japan). Sections were then incubated with 1/200 of anti-α-SMA (Abcam, cat# ab32575, Cambridge, UK) for 1 h, followed by incubation with secondary antibody (HRP-Goat Anti-Rabbit IgG Antibody, Vector laboratories, Burlingame, CA, USA); enzyme-substrate reactions were performed with 3,3′-diaminobenzidine/H_2_O_2_ solution (Nichirei Bioscience Inc., Tokyo, Japan). Bright field images of α-SMA-positive areas were taken around the central vein, with the digital camera DFC295 (Leica, Wetzlar, Germany) at 200× magnification, and the positive areas in 5 fields/section were measured using ImageJ software (National Institute of Health).

### 2.4. Cell Culture and Repopulation of 3D Human Liver Scaffolds

This study was performed at Engitix Limited (London, UK) by using healthy human liver ECM 3D scaffolds developed via proprietary methods [13]. LX-2 cells obtained from Professor Pinzani’s group (University of Florence, Florence, Italy) were cultured and amplified at 37 °C and 5% of CO_2_ in complete Iscove’s Dulbecco’s Medium containing 10% FBS, 1% sodium pyruvate 100×, 1% EM-NEAA 100×, 1% antibiotic-antimycotic 100×, and 1% l-glutamine 200 mM (cIMDM) from Gibco, ThermoFisher Scientific. Cells were sub-passaged 3 times before the experiment.

Prior to cell seeding, human liver tissue was decellularized and sterilized, resulting in 72 decellularized human liver scaffolds that were placed in individual wells with cIMDM for 24 h, as previously described [13]. Briefly, on day 0, LX2 cells were detached with trypsin-EDTA 0.25% phenol red (Gibco, ThermoFisher Scientific), and human liver 3D scaffolds were repopulated with 0.5 × 10^6^ cells/scaffold in a 96-well plate.

On day 1, bioengineered 3D scaffolds were transferred to individual wells in a 48-well plate with cIMDM. On day 12, 3D scaffolds were transferred into new 48-well plates containing cIMDM (*n* = 24) (Control group), cIMDM with 5 ng/mL of TGF-β1 (*n* = 24) (TGF-β1 group), 1 µM NV556 with 5 ng/mL of TGF-β1 (TGF-β1 + NV556 early treatment group) (*n* = 12), or cIMDM with 1 µM NV556 (NV556 early treatment group) (*n* = 12). All groups had replenishment of media with their respective treatments on day 14 and 16.

In addition, on day 14, 12 replicates from the control and TGF-β1 groups were treated with 1 µM of NV556 with (TGF-β1+ NV556 late treatment) or without a new dose of TGF-β1 (NV556 late treatment), depending on the previous conditions. Both treatment groups had replenishment of media on day 16. On day 18, supernatants were collected, and scaffolds were snap-frozen and stored at −80 °C.

For the dose-response effect, cell preparation and activation were performed as previously described, with a deviation from the original protocol in the starting point of the treatment and the exposure time. On day 13, groups were divided depending on the NV556 concentration (0, 0.2, 1, or 5 µM of NV556), with or without 5 ng/mL TGF-β1 in cIMDM, and incubated for 24 h.

### 2.5. Gene Expression

#### 2.5.1. LX-2 Cells

RNA was extracted with the use of an RNeasy Plus Universal mini kit (Qiagen, Hilden, Germany), as previously described by Mazza G. et al. [13]. Bioengineered 3D liver scaffolds were homogenized for 4 min with Qiazol reagent in a TissueLyser, using a 7 mm bead, and the rest of the protocol was followed according to the providers’ instructions. Concentration and purity of the RNA extracted were measured with NANODROP 2000 (ThermoFisher Scientific). RNA integrity was assessed with Bioanalyzer 2100, using an RNA 6000 Nano kit (Agilent, Santa Clara, CA, USA). Reverse transcription was performed with the High-Capacity cDNA Reverse Transcription kit (Applied Biosystems, ThermoFisher Scientific, Foster City, CA, USA), according to their protocol, and run on a Q-cycler II (Quanta Biotech, Byfleet, UK). Real-time PCR was analyzed by using gene-specific TaqMan assays on demand and TaqMan Universal Master Mix (Applied Biosystems, ThermoFisher Scientific) and run in the ABI 7500 Fast Real Time PCR system (Applied Biosystems, ThermoFisher Scientific). The relative expression was quantified through the use of the following TaqMan assays: COL1A1 (TaqMan assay ID: Hs00164004_m1), LOX (TaqMan assay ID: Hs00942483_m1), TGFBI (Hs00998133_M1), TGFBRII (Hs00234253_m1), COL3A1(Hs00943809_m1), COL4A1 (Hs00266237), MMP1 (Hs00899658_m1), CTGF (Hs00170014_m1), and TIMP-4 (Hs00162784_m1). GAPDH (TaqMan assay ID: Hs02786624_g1) was used as a housekeeping gene. Relative quantification was obtained through the use of the ΔΔCt method [32].

#### 2.5.2. STAM Model

For the analysis of LOX, COL1A1, COL3A1, and COL4A1, RNA from liver tissues was extracted and quantified as described above, but with the use of a stainless-steel bead of 5 mm and RNEasy lipid tissue mini kit (Qiagen), following the providers’ instructions. Reverse transcription was performed as described above. Real-time PCR was performed by using SYBR Green PCR Master Mix (Applied Biosystems) and run in QuantStudio 7 flex (Applied Biosystems). The relative expression was quantified through the use of the following primer sequences (Tag Copenhagen) from 5′ to 3′: COL1A1 (forward: CCTCAGGGTATTGCTGGACAAC and reverse: CAGAAGGACCTTGTTTGCCAGG), LOX (forward: CATCGGACTTCTTACCAAGCCG and reverse: GGCATCAAGCAGGTCATAGTGG), COL3A1 (forward: GACCAAAAGGTGATGCTGGACAG and reverse: CAAGACCTCGTGCTCCAGTTAG), and COL4A1 (forward: ATGGCTTGCCTGGAGAGATAGG and reverse: TGGTTGCCCTTTGAGTCCTGGA). GAPDH (forward: CATCACTGCCACCCAGAAGACTG and reverse: ATGCCAGTGAGCTTCCCGTTCAG) was used as a housekeeping gene. Relative quantification was calculated as above.

For the analysis of COL1A2, TIMP-1, and TGFB1, RNA from liver tissue was extracted by using RNAiso (Takara Bio), according to the provider’s instructions. RNA was reverse-transcribed using a reaction mixture containing MgCl2 (F. Hoffmann-La Roche), RNase inhibitor (Toyobo), dNTP (Promega), random hexamer (Promega), 5× first strand buffer (Promega), dithiothreitol (Invitrogen), and MMLV-RT (Invitrogen). Real-time PCR was performed by using real-time PCR DICE and SYBR premix Taq (Takara Bio). The relative expression was quantified by using the following PCR primers from 5′ to 3′: TGFB1 (forward: GTGTGGAGCAACATGTGGAACTCTA and reverse: TTGGTTCAGCCACTGCCGTA), COL1A2 (forward: CCAACAAGCATGTCTGGTTAGGAG and reverse: GCAATGCTGTTCTTGCAGTGGTA), and TIMP-1 (forward: TGAGCCCTGCTCAGCAAAGA and reverse: GAGGACCTGATCCGTCCACAA). Moreover, 36B4 (forward: TTCCAGGCTTTGGGCATCA and reverse: ATGTTCAGCATGTTCAGCAGTGTG) was used as a housekeeping gene. Relative quantification was calculated as above.

### 2.6. ELISA

Supernatants collected from the bioengineered 3D liver scaffolds were analyzed for procollagen type I with procollagen 1 ELISA kit (Abcam, cat#ab210966, Cambridge, UK) and for MMP-2 with MMP-2 ELISA kit (R&D systems, cat#MMP200, Minneapolis, MN, USA), according to the manufacturer’s instructions (Abcam).

For the measurement of SMAD2 and SMAD3, a 7-plex signaling assay for cell lysates was obtained by combining the TGF-β Signaling 6-plex Magnetic bead kit (Merck Millipore, cat# 48-614MAG, Burlington, MA, USA), analyzing Ser465/Ser467 for SMAD2, and Ser423/Ser425 for SMAD3, with a total β-tubulin Magnetic bead MAPmate (Merck Millipore, cat# 46-713MAG) used for normalization.

### 2.7. Data and Statistical Analysis

Data involving two groups were analyzed by a paired T-test, and all data involving more than two groups were analyzed by one-way ANOVA, followed by Dunnett’s or Tukey’s multiple comparison test, or Kruskal Wallis, followed by Dunn’s multiple comparison test **p* < 0.05, ***p* < 0.01 and ****p* < 0.001. Analysis was performed with GraphPad Prism 8.1.1 (San Diego, CA, USA).

## 3. Results

### 3.1. NV556 Decreases Fibrosis in MCD Model

NV556 is a potent sanglifehrin-based cyclophilin inhibitor that was produced by combination of bioengineering and semisynthetic approaches. Oral dosing of NV556 and other similar sanglifehrin-based cyclophilin inhibitors resulted in high liver exposure, likely due to first-pass extraction and high affinity to target cyclophilins in the liver [26], presenting the liver as a suitable organ to study biological effects of the compound. First, the NV556 antifibrotic effect was tested in two well-established nonalcoholic steatohepatitis (NASH) animal models, including the MCD and STAM in vivo model. In the MCD model, mice were fed the methionine choline deficient diet for eight weeks. One week after starting the MCD diet, mice were orally treated with vehicle, 100 mg/kg of NV556, or 30 mg/kg of obeticholic acid. This FXR agonist was used as a reference drug for NASH treatment due to its clinical potency in decreasing steatosis, hepatocellular ballooning, lobular inflammation, and fibrosis [29,33] (Figure 1A).

Before treatment, a loss of 5 g in body weight was observed in MCD-fed mice; this reduction in body weight was observed throughout the treatment period, but no significant changes were observed between groups (Figure 1B). Plasma Aspartate Transaminase (AST) and Alanine Transaminase (ALT) levels (Figure 1C) were significantly reduced at seven weeks, for NV556, in comparison to the vehicle-treated group. NV556-treated animals showed no significant differences in liver weight in relation to the total body weight. In contrast, obeticholic acid treatment showed a significant increase in liver-to-total-body-weight ratio in comparison to the vehicle group (Figure 1D). Analysis of liver total cholesterol resulted in a significant increase in the obeticholic acid treated group in comparison to the vehicle group, but no significant differences were observed for any of the treatments for liver triglycerides and fatty acids (Figure 1E). NV556-treated animals showed no differences in the levels of cholesterol, triglycerides, and fatty acids compared to the controls (Figure 1E). Histological analysis of the collected livers demonstrated no differences in the inflammation score between the groups of animals (Figure 1F). In the obeticholic-acid-treated group, a significant reduction in liver steatosis and ballooning was detected, but no significant effect in these parameters was observed for the NV556 treated group (Figure 1F). The percentages of Sirius red-positive areas were significantly reduced in NV556 compared to the vehicle or obeticholic-acid-related groups (Figure 1G). These results demonstrate that NV556 significantly attenuated the increase in liver transaminases and reduced the degree of liver fibrosis in the MCD model.

### 3.2. NV556 Decreases Fibrosis in STAM Model

Next, we investigated the effect of NV556 in the STAM model of nonalcoholic steatohepatitis, induced by injecting animals with 200 µg of streptozotocin, an antibiotic cytotoxic to pancreatic β-cells, followed by a high-fat diet (HFD). Telmisartan, which is an angiotensin II receptor antagonist and peroxisome proliferator-activated receptor-γ agonist, has the capacity to ameliorate NASH by decreasing fibrosis and inflammation [34]. In our STAM model, Telmisartan was used as a reference drug. Mice were orally treated with NV556 at a concentration of 100 mg/kg, or Telmisartan at a concentration of 5 mg/kg (Figure 2A). No significant effect on body weight was observed between the treatment and the vehicle groups (Figure 2B). At the end of the experiment, NV556-treated animals showed no differences in liver to total body weight compared to the vehicle-treated mice (Figure 2C). As expected, Telmisartan reduced liver to body weight in this group of animals (Figure 2C). Analyzing whole blood glucose demonstrated a decrease for the NV556 treatment group at six weeks, which was absent at seven weeks in comparison to the vehicle group. No significant differences were observed for plasma triglyceride and plasma insulin between the vehicle and any of the treatment groups (Figure 2D). At 12 weeks, liver triglyceride content was measured for each group, indicating a significant decrease for the Telmisartan-treated group, but no changes were observed for the NV556 treated group (Figure 2E).

Liver sections were assessed for micro- and macro-vesicular fat deposition, hepatocellular ballooning, and inflammatory cell infiltration, with no changes between different groups except for a significant decrease in ballooning with Telmisartan treatment (Figure 2F). Similar to the results obtained in the MCD model, treatment of mice with NV556 in the STAM model resulted in a significant decrease in Sirius red staining compared to the vehicle group, indicating a NV556-induced reduction in collagen deposition (Figure 2G). This result demonstrates that NV556 significantly reduced the degree of liver fibrosis in the STAM model, without affecting body weight.

### 3.3. NV556 Treatment Reduces Collagen I Production in LX2 Human Hepatic Stellate Cells

Since NV556 significantly reduced collagen deposition in the experimental STAM and MCD in vivo models and hepatic stellate cells are a key player in collagen deposition, we analyzed NV556 antifibrotic effect in a 3D model, using LX2 cells, a well-characterized human hepatic stellate cell line engrafting 3D liver ECM scaffolds [13]. When examining the effect of NV556 on non-stimulated LX2 cells, a significant decrease in mRNA expression of lysyl oxidase (LOX) and collagen type I alpha I chain (COL1A1) was observed in both the NV556-early and -late treatment group (Figure 3A,B).

In addition, a significant decrease in procollagen I secretion was detected in comparison to the control group (Figure 3C). When 3D ECM scaffolds repopulated with LX2 cells were activated by treatment with TGF-β1, the activation was confirmed by analysis of HSC-activation markers (Appendix A). NV556 treatment reduced COL1A1 and LOX in the TGF-β1+NV556-early and -late treatment groups in comparison to the TGF-β1 group (Figure 4A,B), similar to the data observed in non-stimulated LX2. In addition, ELISA analysis confirmed a decrease in secretion of procollagen I when cells were treated with NV556 (Figure 4C).

When different concentrations of NV556 (0.2, 1.0, and 5.0 µM) were tested on activated LX2 cells in the 3D scaffold model, a significant decrease in LOX and COL1A1 expression could be observed at the lowest concentrations of NV556 (Figure 5A,B). Furthermore, collagen IV (COL4A1) showed similar reduced gene expression at the two highest NV556 concentrations tested, while collagen 3 (COL3A1) was unaffected (Figure 5C,D). Expression of several genes involved in HSC activation were analyzed, but no significant changes were observed for CTGF, TGFB1, and TGFBRII (Appendix A). These results suggest that NV556 treatment selectively interferes with COL1A1 and LOX transcription, resulting in reduced collagen production in LX2 cells.

To confirm these data, liver samples from the STAM model were further analyzed for gene-expression profiling of COL1A1, COL1A2, COL3A1, COL4A1, LOX, TIMP-1, and TGFB1, and IHC staining of α-SMA. No changes in the levels of any of the genes or percentage of α-SMA-positive areas were detected, when comparing the vehicle group to any of the treatment groups (Appendix A).

## 4. Discussion

Chronic liver diseases, a major cause of morbidity and mortality worldwide, are characterized by chronic inflammation and fibrogenesis, resulting in liver cirrhosis and life-threatening complications. The potential therapeutic strategies to develop antifibrotics include prevention of hepatic inflammation, inhibiting the activation of HSC, and intervening in ECM production/degradation. In the present study, we demonstrated the potential of the sanglifehrin-based cyclophilin inhibitor NV556 as an antifibrotic compound, which acts by reducing the production of the extracellular matrix collagens.

Previous studies identified the role of cyclophilin A as a proinflammatory signal by being secreted to the extracellular environment in response to inflammation and cell death [35]. In a rodent model of CCl_4_-induced liver fibrosis, exposure of animals to the cyclosporin derivative NIM811 resulted in reduced liver inflammation [18]. Treatment with the cell-impermeable cyclophilin inhibitor MM284 in an animal model of autoimmune myocarditis reduced myocardial inflammation by lowering the recruitment of T cells and macrophages [19]. In the two fibrotic models examined in the present study, we demonstrated no significant effect of cyclophilin inhibitor NV556 on lobular inflammation.

Cyclophilin D, a key regulator in the mitochondrial permeability transition pore or MPTP, has been implicated in tubular cell apoptosis and interstitial fibrosis in the obstructed kidney in vivo model [36]. Treatment with NIM811 demonstrated a protective role in hepatocytes by reducing liver necrosis in animal models of fibrosis due to bile duct ligation (BDL), and CCl_4_ treatment and was suggested to act via inhibition of the mitochondrial permeability transition [18,24]. Furthermore, NIM811 delayed acetaminophen-induced necrotic cell death in hepatocytes [21]. In the CCl_4_-induced liver fibrosis rat model, a significant decrease of liver fibrosis and serum levels of ALT and AST was observed without any significant changes on liver steatosis [18]. In our study, NV556 treatment of mice demonstrated no major effect on ballooning or steatosis, including quantification of liver triglycerides in either the STAM or MCD model. Other measurements such as plasma triglyceride content in the STAM model or liver total cholesterol, triglyceride, or fatty acids in the MCD model did not present significant differences between vehicle and NV556 treatment groups. On the other hand, a significant decrease in AST and ALT levels was observed in NV556-treated mice in comparison to the vehicle control groups, suggesting a protective effect of NV556 on the liver, acting downstream of inflammation and ballooning; and it is compatible with an inhibitory effect on cyclophilin D.

Analysis the NAFLD parameters in both the STAM and MCD models demonstrated a significant reduction in liver fibrosis, measured by Sirius-red staining after treatment with NV556, as has also recently been observed in an independent setup of the STAM model [37]. Due to the lack of differences between vehicle and NV556 treatment groups on NAFLD activity markers, such as steatosis, inflammation, and ballooning, but a significant decrease in collagen deposition, we hypothesized that NV556 is effective in specific stages of the development of fibrosis, such as ECM deposition. Since hepatic stellate cells play a major role in collagen deposition, the effect of NV556 was assessed in a 3D human liver scaffold model that was bioengineered with hepatic stellate LX2 cells. In this 3D in vitro model, a downregulation in mRNA expression of COL1A1 and secretion of procollagen I was observed in cells treated with NV556, with or without TGF-β1. Importantly, NV556 titration caused a significant decrease in COL1A1 expression, already at the lowest concentration used. Similar results on collagen production have been observed with NIM811 treatment due to TGF-β pathway inhibition in HSC [20]. Knock-down of CypB and D was previously shown to decrease collagen type III expression in vitro [18]. In our study, collagen type IV mRNA levels were reduced by NV556 treatment at the two highest concentrations, while collagen type III was unaffected. Additionally, CypB has been presented as a key factor for the correct folding of collagen type I, and mutations in the CypB gene PPIB have been linked to the occurrence of Osteogenesis Imperfecta in humans [38]. In addition, PPIB KO mice presented a slow speed folding collagen type I and over-modification of lysyl residues, resulting in skeletal abnormalities [39]. Moreover, CsA exposure on cultured fibroblasts resulted in an increased accumulation of post-translational modifications of collagen I and III and increased intracellular degradation of these two types [40]. In this study, we did not explore the NV556-induced post-translational modifications in activated hepatic stellate cells, but, given the importance of CypB for the correct collagen folding, we speculate that this could be one of the mechanisms for the observed decreased secretion of procollagen type I.

In the 3D liver scaffold model, in addition to reduced expression of collagen, a significant reduction in LOX gene expression could be observed. LOX, which is a marker of activated hepatic stellate cells, is involved in cross-linking of collagens and elastin, contributing to the stiffness of the extracellular matrix [4,41,42]. LOX inhibition has been shown to reduce the accumulation of cross-linked collagens in vivo [41]. Therefore, the significant decrease in LOX production observed in our study for both NV556-early and -late treatment group at a low concentration, could represent an additional mechanism contributing to the antifibrotic effect observed in our in vivo NASH models.

Previous publications have demonstrated the importance of cyclophilins, such as CypA or CypB, for the activation of HSC through the TGF-β pathway, where inhibition of CypB or extracellular CypA suppressed the phosphorylation of SMAD2 and SMAD3 and increased SMAD7 expression, thus inhibiting the TGF-β1 pathway [20] and leading to an increase in collagenase activity [17,18]. In our 3D liver scaffold model, we could not find significant changes in several markers for TGF-β activation in LX2, such as CTGF or TGFB1, following NV556 treatment.

A gene expression profiling of COL1A1, COL1A2, COL3A1, COL4A1, LOX, TIMP-1, and TGFB1 did not present significant changes in the levels of any of the genes or α-SMA-positive area, when comparing the vehicle control with NV556 or telmisartan-treated animals. These results differed from the in vitro experiments, which demonstrated significant and robust changes in the levels of COL1A1 and LOX (Figure 3, Figure 4 and Figure 5) under various conditions. The reason behind this discrepancy could potentially be related to the differences in experimental setups between the in vitro and in vivo experiments. The gene expression profile was analyzed within days in the in vitro experiment, whereas samples from the in vivo study were taken at the study endpoint at 12 weeks, where the liver fibrosis was already established in all groups.

Taken together, our data suggest that NV556 can be considered a candidate drug for treatment of liver fibrosis. The antifibrotic effects of NV556 seem to be mediated through a reduction in the extracellular matrix gene expression, such as collagen 1A1, and by interfering with the cross-linking of collagens and elastin by hepatic stellate cells.

## Figures and Tables

**Figure 1 cells-08-01409-f001:**
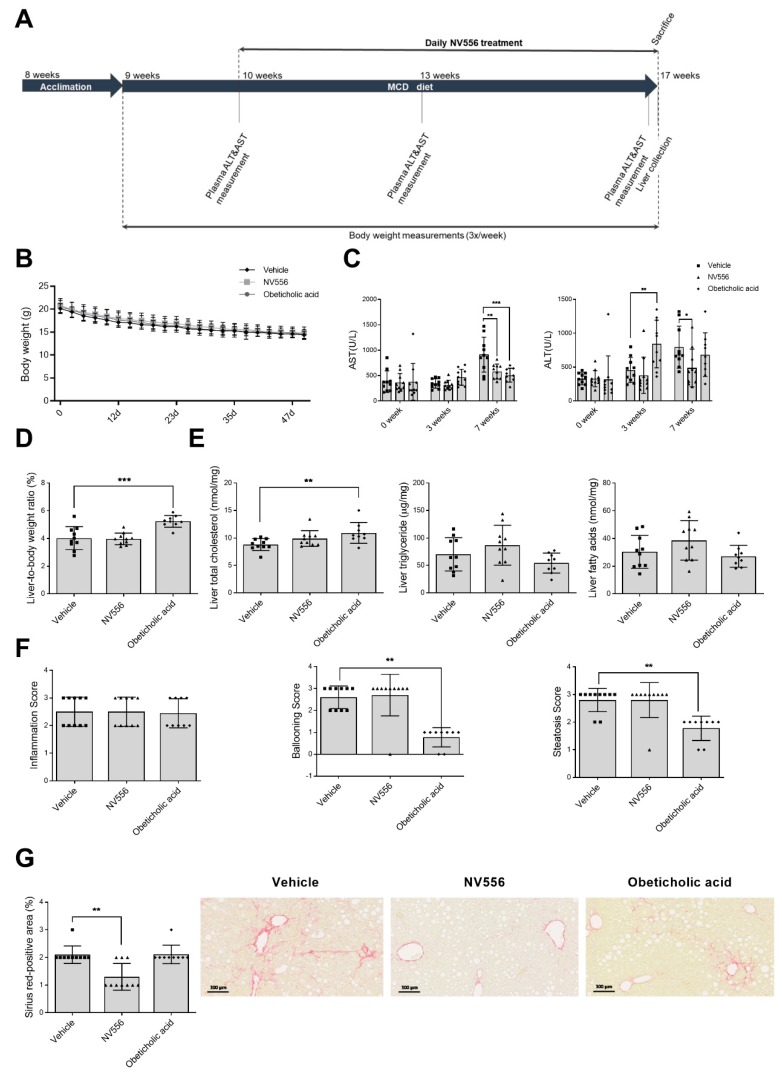
Effect of NV556 treatment in methionine-choline-deficient (MCD)-diet-fed mice. (**A**) Experimental timeline and parameters measured: (**B**) body weight, (**C**) AST, and ALT, (**D**) liver/body weight ratio (%), (**E**) liver total cholesterol, liver triglycerides, and liver fatty acids; and (**F**) NAFLD scoring–inflammation, ballooning, steatosis score, and (**G**) fibrosis stained by Sirius red. Representative photomicrographs of Sirius red staining (Scale bars indicate 100 µm). Data are represented as mean ± SD and statistically analyzed by one-way ANOVA, followed by Dunnett’s multiple comparison test for **C**, **D**, and **E**. **F** and **G** were statistically analyzed by Kruskal Wallis, followed by Dunn’s multiple comparison test, **p* < 0.05, ***p* < 0.01, ****p* < 0.001, *n* = 10 mice per treatment.

**Figure 2 cells-08-01409-f002:**
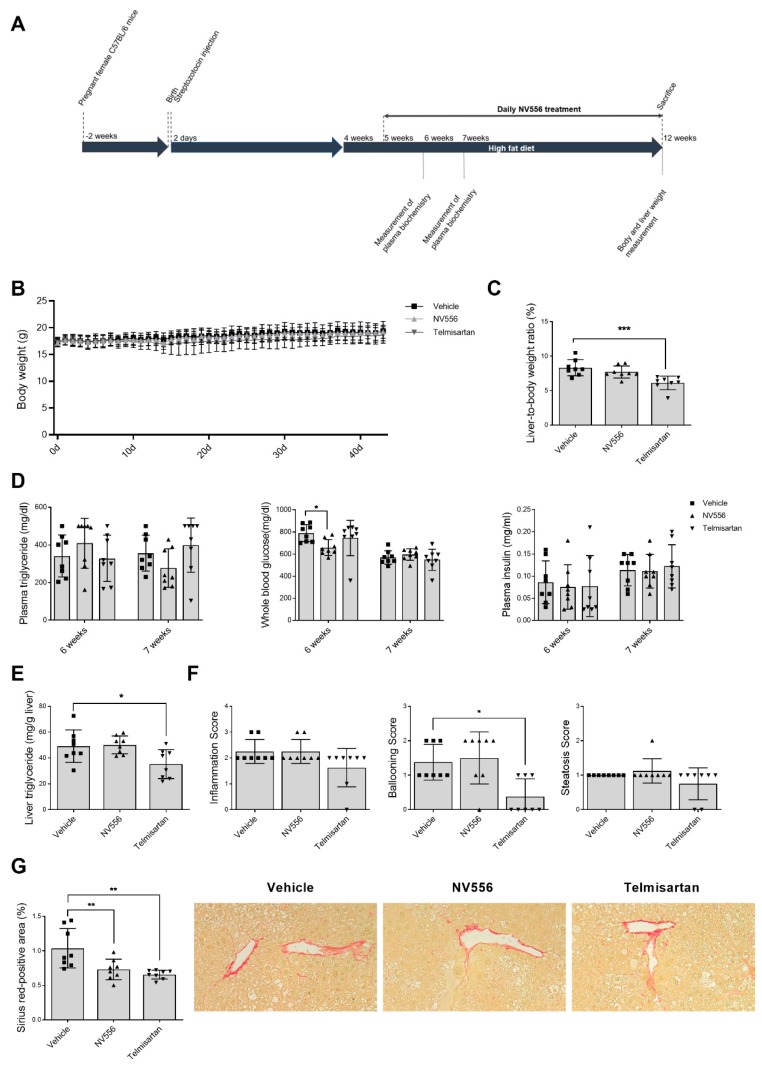
Effect of NV556 treatment in the STAM model of nonalcoholic steatohepatitis (**A**) Experimental timeline and parameters measured: (**B**) body weight and (**C**) liver/body-weight ratio, (**D**) plasma triglyceride, whole blood glucose, plasma insulin, (**E**) liver triglyceride, and (**F**) NAFLD scoring: inflammation, ballooning, and steatosis score (**G**) fibrosis area: representative photomicrographs at 200× and values of Sirius-red positive areas. Data are represented as mean ± SD and statistically analyzed by one-way ANOVA, followed by Dunnett’s multiple comparison test for C, D, E, and G. F was statistically analyzed by Kruskal–Wallis, followed by Dunn’s multiple comparison test **p* < 0.05, ***p* < 0.01, ****p* < 0.001, *n* = 8 mice per treatment.

**Figure 3 cells-08-01409-f003:**
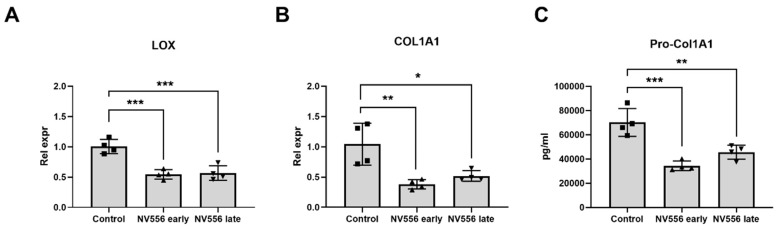
NV556 effect on gene expression (**A**,**B**) and procollagen secretion (**C**) in a 3D human liver model reseeded with LX2 cells. Data in **A** and **B** are represented as mean of relative expression over nontreated control cells ± SD, and data in **A**–**C** are statistically analyzed by one-way ANOVA, followed by Tukey’s multiple comparison test. **p* < 0.05, ***p* < 0.01 and ****p* < 0.001, *n* = 4 scaffolds per condition investigated.

**Figure 4 cells-08-01409-f004:**
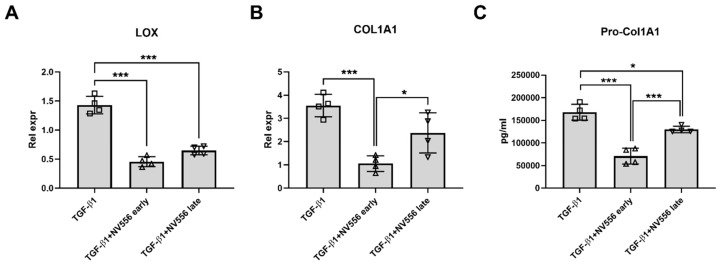
NV556 effect on gene expression (**A**,**B**) and procollagen secretion (**C**) in a 3D human liver model reseeded with LX2 cells. Data in **A** and **B** are represented as mean of relative expression over nontreated control cells ± SD and Data in **A**–**C** are statistically analyzed by one-way ANOVA, followed by Tukey’s multiple comparison test. **p* < 0.05, ***p* < 0.01 and ****p* < 0.001, *n* = 4 scaffolds per condition investigated.

**Figure 5 cells-08-01409-f005:**
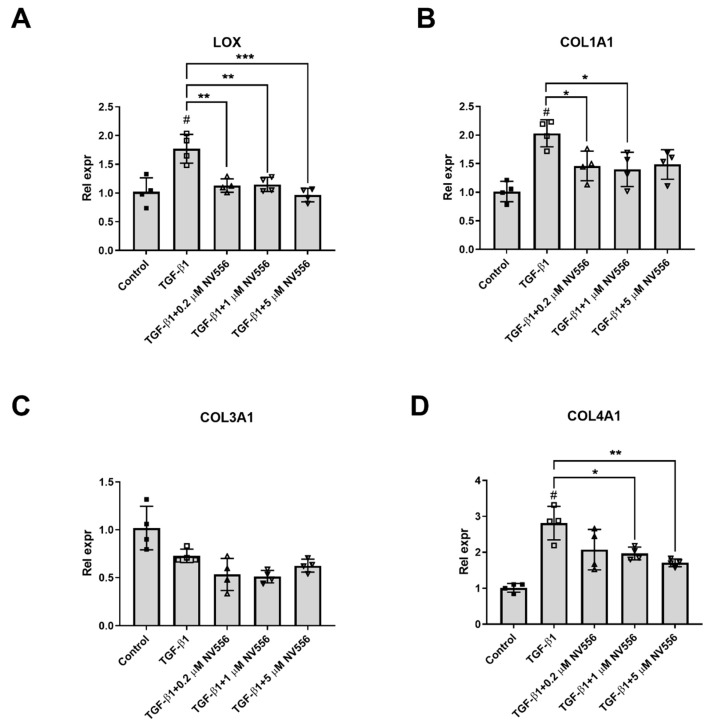
NV556 dose-response effect on gene expression in 3D human liver model reseeded with LX2 cells. Data are represented as mean of relative expression to inactive control ± SD for LOX (**A**), COL1A1 (**B**), COL3A1 (**C**) and COL4A1 (**D**) expression and statistically analyzed by one-way ANOVA, followed by Tukey’s multiple comparison test. **p* < 0.05, ***p* < 0.01, and ****p* < 0.001. #*p* <0.05 for TGF-β1 versus nontreated Control cells, *n* = 4 scaffolds per condition investigated.

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
