# Peer review of "Evaluation of NV556, a Novel Cyclophilin Inhibitor, as a Potential Antifibrotic Compound for Liver Fibrosis"

_cells, 2019, doi:10.3390/cells8111409_

Round 1
Reviewer 1 Report
In this study "Evaluation of NV556, a novel cyclophilin inhibitor, as a potential antifibrotic compound for liver fibrosis", authors investigated the potential of the sanglifehrin-based cyclophilin inhibitor NV556 as an anti-fibrotic compound acting by reducing the production of the extracellular matrix collagens. The manuscript is clearly and well written. However, this study has several limitations.
1) The anti-fibrotic effect was demonstrated by quantification of Sirius red staining only. To support anti-fibrotic activity, qPCR analysis of alpha smooth muscle actin (α-SMA), Collagen (COL)1 and possibly also LOX, TIMP1, MMP2, MMP9 and TGFβ gene expression in liver tissue of both models can be added.
2) Why the most classical CCl4 induced model of liver fibrosis was not used to test anti-fibrotic effect of NV556 ?
3) The novelty is limited as authors published recently the message about NV556 anti-fibrotic activity elsewhere, https://doi.org/10.3389/fphar.2019.01129 , "Cyclophilin Inhibitor NV556 Reduces Fibrosis and Hepatocellular Carcinoma Development in Mice With Non-Alcoholic Steatohepatitis"
4) Concerning STAM model, did you observed tumors in 12 week old mice?
5) I don't understand the rationale behind using Telmisartan (treatment of hypertension) in STAM model. Why authors did not used Obeticholic Acid as a reference drug, like in MCD model?
6) Is it possible to provide ALT and AST levels in STAM model?
7) Figure 2A - Collection of fecal pellets is highlighted. But there are no related analysis/data in manuscript.
Author Response
In this study "Evaluation of NV556, a novel cyclophilin inhibitor, as a potential antifibrotic compound for liver fibrosis", authors investigated the potential of the sanglifehrin-based cyclophilin inhibitor NV556 as an anti-fibrotic compound acting by reducing the production of the extracellular matrix collagens. The manuscript is clearly and well written. However, this study has several limitations.
Reply: We want to thank the reviewer for the careful review of our manuscript and the valuable and constructive suggestions.
The anti-fibrotic effect was demonstrated by quantification of Sirius red staining only. To support anti-fibrotic activity, qPCR analysis of alpha smooth muscle actin (α-SMA), Collagen (COL)1 and possibly also LOX, TIMP1, MMP2, MMP9 and TGFβ gene expression in liver tissue of both models can be added.
Reply: We have now included additional information regarding gene expression profiling of COL1A1, COL1A2, COL3A1, COL4A1, LOX, TIMP-1 and TGFB1 as well as the percentage of α-SMA IHC staining positive area in STAM model (Suppl. Fig. 3). No changes in the levels of any of the genes or α-SMA positive area could be detected comparing vehicle control with NV556-treated animals. Nor were there any significant effects in the positive control group treated with Telmisartan. These results differed to the in vitro experiments, which demonstrated significant and robust changes in the levels of COL1 and LOX (Fig. 3-5) under various conditions. The reason behind this discrepancy could potentially be related to the differences in experimental setups between the in vitro and in vivo experiments. The gene expression profile was analyzed within days in the in vitro experiment, whereas samples from the in vivo study were taken at the study endpoint at 12 weeks, where the liver fibrosis was already established in all groups. We have commented these issues in the manuscript. Unfortunately, we do not have access to additional samples from the MCD model.
2) Why the most classical CCl4 induced model of liver fibrosis was not used to test anti-fibrotic effect of NV556?
Reply: Development of fibrosis in the CCl4-model is rapidly progressing following a chemically induced insult (https://www.ncbi.nlm.nih.gov/pmc/articles/PMC4542084/). In our study, we aimed to perform a prolonged exposure of NV556 in metabolic models to monitor the effect of NV556 in a more chronic progression of the disease.
3) The novelty is limited as authors published recently the message about NV556 anti-fibrotic activity elsewhere, https://doi.org/10.3389/fphar.2019.01129 , "Cyclophilin Inhibitor NV556 Reduces Fibrosis and Hepatocellular Carcinoma Development in Mice With Non-Alcoholic Steatohepatitis"
Reply: The recent published study by Kuo et al., 2019, presented a general in vivo effect of NV556 in NASH and NASH-induced hepatocellular carcinoma. We have included this paper in the reference list of our manuscript. In the present study we show confirmatory data from the STAM model and extend these observations to the MCD model. These two models demonstrate a broad anti-fibrotic effect of NV556. In addition, in the present study, we have elucidated mechanistic aspects of the pro-fibrotic process by investigating the effect of NV556 on HSC in a 3D liver ECM scaffold in vitro model.
4) Concerning STAM model, did you observed tumors in 12 week old mice?
Reply: We did not perform any analyses of tumors in the 12-week-old mice. Generally, in the STAM model protocol that we followed, the tumors become visible between weeks 16 and 20.
5) I don't understand the rationale behind using Telmisartan (treatment of hypertension) in STAM model. Why authors did not used Obeticholic Acid as a reference drug, like in MCD model?
Reply: For the STAM model, we collaborated with SMC Laboratories, Inc. Previous studies using Telmisartan in STAM model showed that mice sacrificed at 12 weeks of age had a significant decrease in liver weight, decreased NAFLD activity score including steatosis, inflammation and ballooning score as well as significant decrease in the Sirius red staining positive areas (unpublished studies, SMC Laboratories, Inc). A recent publication also found the positive effects of Telmisartan in the progression of NASH (Connectivity mapping of angiotensin-PPAR interactions involved in the amelioration of non-alcoholic steatohepatitis by Telmisartan).
For the MCD model, we collaborated with Physiogenex S.A.S. Earlier studies confirmed the efficacy of the farnesoid X nuclear ligand obeticholic acid in NASH development. This treatment reduced liver inflammation and presented a tendency to decrease the NAFLD score in hamster (https://www.sciencedirect.com/science/article/pii/S0014299917307549?via%3Dihub). In addition, the clinical effect of obeticholic acid in NASH where all parameters of the NAFLD activity score and fibrosis were improved in most of the patients but with an increased total serum cholesterol (https://www.sciencedirect.com/science/article/pii/S0140673614619334?via%3Dihub).
We have included additional information about Telmisartan and Obeticholic acid in the text.
6) Is it possible to provide ALT and AST levels in STAM model?
Reply: Unfortunately, ALT and AST were not measured at the time of the study and there are no remaining samples to be analyzed.
7) Figure 2A - Collection of fecal pellets is highlighted. But there are no related analysis/data in manuscript.
Reply: Fecal pellets were collected for a potential additional testing. As no further tests were performed we have now removed “collection of fecal pellets” from the manuscript.
Reviewer 2 Report
The present manuscript aimed to investigate the antifibrotic effects of NV556, a new potent sanglifehrin-based cyclophilin inhibitor in vivo and in vitro. The Authors showed that NV556 decreased liver fibrosis in MCD and STAM in vivo models and reduced collagen production in TGFβ1-activated hepatic stellate cells in vitro. The paper is very interesting but the Authors need to improve the following points:
Regarding the methionine choline deficient diet (MCD) in vivo model, the Authors should remember better the known effects of the treatment, together with the role of the obeticholic acid In the results section, they should explain better why NV556 did not decrease the levels of cholesterol, triglycerides and fatty acids compared to the control model. Why in the STAM model, they did not consider the levels of transaminases, cholesterol and fatty acids? In addition in the same STAM model, the Authors should remember and explain the role of Telminsartan. In addition to the Sirius Red, the Authors could add the analysis and quantification of α-SMA to verify the activation in vivo of the hepatic stellate cells. It could be an important passage and link between the in vivo and the in vitro study about the role of HSC in NV556 treatment.Author Response
The present manuscript aimed to investigate the antifibrotic effects of NV556, a new potent sanglifehrin-based cyclophilin inhibitor in vivo and in vitro. The Authors showed that NV556 decreased liver fibrosis in MCD and STAM in vivo models and reduced collagen production in TGFβ1-activated hepatic stellate cells in vitro. The paper is very interesting but the Authors need to improve the following points:
Reply: We want to thank the reviewer for the careful review of our manuscript and the valuable and constructive suggestions.
Regarding the methionine choline deficient diet (MCD) in vivo model, the Authors should remember better the known effects of the treatment, together with the role of the obeticholic acid.
Reply: A description regarding the clinical evidence of obeticholic acid in NASH has been added in the manuscript.
In the results section, they should explain better why NV556 did not decrease the levels of cholesterol, triglycerides and fatty acids compared to the control model.
Reply: We have included a more detailed explanation in the discussion.
Why in the STAM model, they did not consider the levels of transaminases, cholesterol and fatty acids?
Reply: These were unfortunately not measured at the time of the study, and there are no remaining samples to be analyzed.
In addition, in the same STAM model, the Authors should remember and explain the role of Telminsartan.
Reply: A description regarding the effects of Telmisartan in NASH progression has been added to the manuscript.
In addition to the Sirius Red, the Authors could add the analysis and quantification of α-SMA to verify the activation in vivo of the hepatic stellate cells.
Reply: We included additional information regarding the percentage of α-SMA IHC staining positive area in STAM model (Suppl. Fig. 3). No changes in the levels α-SMA positive area could be detected comparing vehicle control with NV556-treated animals. These results differed to the in vitro experiments, which demonstrated significant and robust changes in the levels of COL1 and LOX (Fig. 3-5) under various conditions. The reason behind this discrepancy could potentially be related to the differences in experimental setups between the in vitro and in vivo experiments. The gene expression profile was analyzed within days in the in vitro experiment, whereas samples from the in vivo study were taken at the study endpoint at 12 weeks, where the liver fibrosis was already established in all groups. We have commented these issues in the manuscript.
It could be an important passage and link between the in vivo and the in vitro study about the role of HSC in NV556 treatment.
Reply: The link between in vitro and in vivo data has been highlighted in the discussion part of the manuscript.
Round 2
Reviewer 1 Report
I recommend this manuscript for publication.
Reviewer 2 Report
In my opinion the paper is ready for publication.